# Effect of acupuncture or moxibustion at Acupoints Weizhong (BL40) or Chize (LU5) on the change in lumbar temperature in healthy adults: A study protocol for a randomized controlled trial with a 2 × 2 factorial design

**Siyi Zheng**[1], **Qiongying Shen**[1], **Zhengyi Lyu**[1], **Shuxin Tian**[1], **Xiaoxiao Huang**[1], **Yiyue Liu**[1], **Xiaoshuai Yu**[1], **Wei Pan**[1], **Na Nie**[2], **Yi Liang**[1,2]*, **Jianqiao Fang**[1,2]

1 The Third School of Clinical Medicine, Zhejiang Chinese Medical University, Hangzhou, China,
2 Department of Acupuncture and Moxibustion, The Third Affiliated Hospital of Zhejiang Chinese Medical University (Zhongshan Hospital of Zhejiang Province), Hangzhou, China

* liangyiwww@126.com

**Data Availability Statement:** Deidentified research data will be made publicly available when the study

## Abstract

### Background

Low back pain is a common complaint among adults, and moxibustion and acupuncture are commonly used treatments. In traditional theory, Weizhong (BL40) is a popular acupoint, as supported by the saying "Yao Bei Wei Zhong Qiu." However, the difference in efficacy between acupuncture and moxibustion remains unclear. Therefore, this trial will compare the thermal effects of acupuncture and moxibustion at BL40 and Chize point (LU5) in healthy adults to provide more objective evidence regarding the relationship between the lumbar and BL40.

### Method/Design

The trial will use a two-by-two factorial design, randomly assigning 140 participants to four groups (acupuncture at Weizhong (BL40), acupuncture at Chize (LU5), moxibustion at Weizhong (BL40), and moxibustion at Chize (LU5)) at a ratio of 1:1:1:1. Each group will undergo a 30-minute intervention, with the primary outcome being mean temperature in the lumbar region at the last minute of the intervention period. Secondary outcomes include maximum lumbar temperature in the lumbar region at the last minute of the intervention, average lumbar temperature and average bladder meridian temperature at specific time points during and after the intervention, and scores on the warming sensation questionnaire. Data will be analyzed on an intention-to-treat basis.

### Discussion

This study will be the first to compare the thermal effect difference in the lumbar area between acupuncture and moxibustion in healthy individuals. The findings of this study will provide new insights for the "Yao Bei Wei Zhong Qiu" theory of traditional Chinese medicine.

is completed and published. All relevant data from this study will be made available upon study completion. And these data are available from the ResMan research manager repository http://www.medresman.org (accession number(s) zsy, 123456).

**Funding:** This study is funded by The National Key Research and Development Program of China (No. 2018YFC1704600) for JF and YL, and by Zhejiang Chinese Medical University 2023 Graduate Top Innovative Talent Cultivation Program (741100G00726) for SZ and YL. https://www.cncbd.org.cn/Index/ The funders did not and will not have a role in study design, data collection and analysis, decision to publish, or preparation of the manuscript.

**Competing interests:** The authors have declared that no competing interests exist.

**Abbreviations:** BL40, Weizhong; LU5, Chize; CONSORT, Consolidated Standards of Reporting Trials; SPIRIT, Standard Protocol Items: Recommendations for Interventional Trials; CRFs, case report forms; PL, project leader; RA, research assistan; AEs, adverse events; ITT, intention-to-treat; LBP, Low Back Pain.

## Trial registration

ClinicalTrials.gov, Trial number: NCT05665426. Registered on 26 December 2022.

## Background and objectives

Acupuncture and moxibustion are historically significant components of traditional Chinese medicine, and they have been widely accepted and used both domestically and internationally [1]. Moreover, acupuncture has been found to be an effective treatment for various medical conditions, symptoms, and diseases, including low back pain (LBP) [2]. According to the traditional theory of acupuncture and moxibustion, the acupoint Weizhong (BL40) has been widely employed to treat LBP since it was first mentioned in the ancient text *Inner Classic of the Yellow Emperor* (Huang Di Nei Jing, 475 B.C.-221 B.C.). This text marks the earliest recognition of the relationship between BL40 and the lumbar region. In 1439, Xu Feng's *Complete Compendium of Acupuncture and Moxibustion* (Zhen Jiu Da Quan) introduced the guiding statement "Yao Bei Wei Zhong Qiu," which is still used to guide the use of BL40 for the treatment of LBP. Currently, BL40 is a commonly utilized acupoint for the treatment of LBP [3].

In the clinic, changes in skin microcirculation can be visualized directly by the laser Doppler technique (skin blood flow) and indirectly by infrared thermography (body surface temperature). Infrared thermographic imaging is also a reliable and easy-to-handle tool for distinguishing between different interventions of needling [4]. Acupuncture at BL40 can relieve pain in LBP patients, and an increase in skin microcirculation can also be observed in the lumbar region of LBP patients who are treated by needling at BL40 [5, 6], which suggests that the increase in skin microcirculation might be associated with the relief of LBP. Moreover, a similar phenomenon (increase in skin microcirculation after needling at BL40) has also been observed in healthy individuals [7, 8].

Although both acupuncture and moxibustion are widely used in clinical treatment, moxibustion is a less painful, more user-friendly and more widely tolerated approach. However, few studies have compared acupuncture and moxibustion, and the studies that have compared these approaches are heterogenous with respect to the definitions of the observation area, thereby limiting the comprehensive analysis of BL40. To thoroughly evaluate the properties of BL40, a multifactor randomized controlled study should be performed to explore the differences in thermal effects produced by different acupoints and interventions.

Therefore, we will utilize infrared thermography to assess changes in skin microcirculation in the lumbar region by measuring its surface temperature distribution and variations. Due to the anatomical resemblance between BL40 and LU5, LU5 will be selected as the control, and acupuncture and moxibustion will be employed as therapies. In this trial, 140 participants will be recruited and then allocated randomly into four treatment groups: acupuncture at BL40 (Acu-BL40), moxibustion at BL40 (Mox-BL40), acupuncture at LU5 (Acu-LU5), and moxibustion at LU5 (Mox-LU5). The aims of this study are (1) to observe the thermal effect of BL40 in the lumbar area and (2) to compare the thermal effects of acupuncture and moxibustion at BL40.

## Method/Design

This trial will be a randomized, controlled, single-center clinical trial with a 2×2 factorial design. The trial will be conducted at the Third Affiliated Hospital of Zhejiang Chinese Medical University, located in Hangzhou, Zhejiang Province, China. A total of 140 participants will

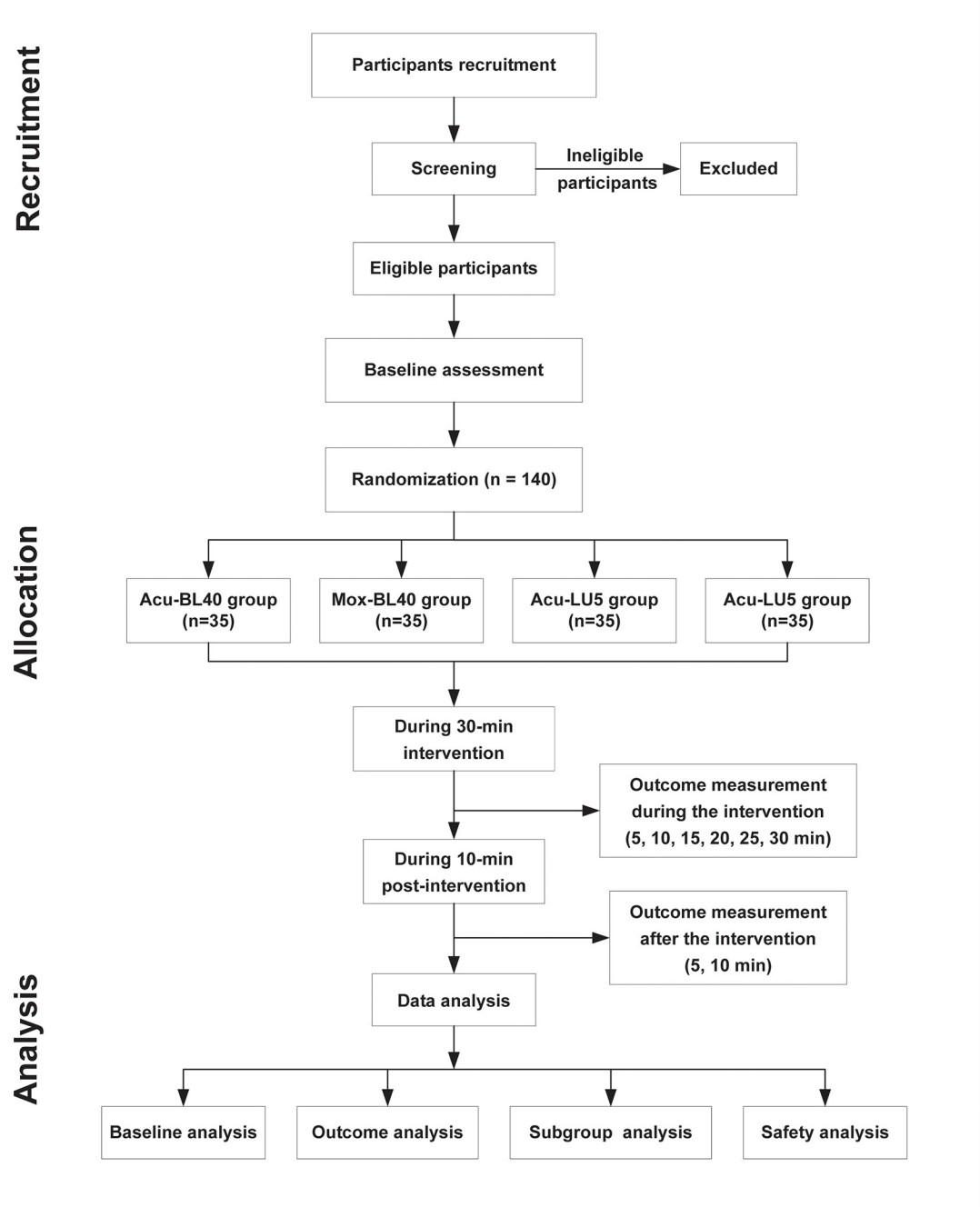

**Fig 1. Flow diagram of participants.** Acu-BL40: Acupuncture at BL40, Mox-BL40: Moxibustion at BL40, Acu-LU5: Acupuncture at LU5, Mox-LU5: Moxibustion at LU5.

be randomly allocated to one of four groups (Acu-BL40, Acu-LU5, Mox-BL40, and Mox-LU5) at a 1:1:1:1 ratio. Fig 1 depicts the flow diagram for this trial.

## Sample size

We estimated the sample size based on mean temperature in the lumbar region at the last minute of the intervention period. As the primary outcome measure. Sample size calculations are

performed to determine the number of participants needed to detect effect sizes. According to previously published results [6, 8] and our preliminary data, the mean average temperature difference are $\mu_1 = 0.84$ $\mu_2 = 0.34$, $\mu_3 = 0.70$, and $\mu_4 = 0.20$ in the Acu-BL40 group, Acu-LU5 group, Mox-BL40 group and Mox-LU5 group, respectively, and the SD $\sigma = 0.26$. k represents the number of groups and equals 4 in this trial. We will assume a type I error of 5% ($\alpha = 0.05$) and 80% power ($\beta = 0.20$). The following formula was used to determine the number of people in each group:

$$N = \lambda \Big/ \frac{1}{\sigma^2} \sum_{i=1}^{k} (\bar{X}_i - \bar{X}_0)^2$$

Accounting for a 20% dropout rate during the study, a total of 140 participants are required to obtain the target of 35 participants per group.

## Randomization, allocation and blinding

The eligible patients will be given a sequential study number by research staff. Subsequently, their basic information will be entered into an internet-based and password-protected clinical research management system (ResMan Research Manager, supported by West China Hospital, Sichuan University, China), which will automatically generate a research code and randomly allocate participants to one of four groups at a ratio of 1:1:1:1: (1) Acu-BL40 group: acupuncture at BL40; (2) Acu-LU5 group: acupuncture at LU5; (3) Mox-BL40 group: moxibustion at BL40 and (4) Mox-LU5 group: moxibustion at LU5. The groups and intervention allocation are listed in Table 2. Given the nature of the intervention, the allocation status will be known to both participants and acupuncturists. However, those responsible for data collection, statistical analysis and interpretation will be blinded to the group allocation. Treatment records will be documented in a dedicated electronic case report form (eCRF) and paper CRF form.

## Participant and recruitment

We will recruit 140 healthy volunteers aged 18–60 through advertising in community centers, WeChat software, and participant referrals. Those who are interested can contact the principal investigator (PI) by phone or email. All eligible participants will be screened in accordance with the inclusion and exclusion criteria. A CONSORT diagram of participant recruitment and SPIRIT figure are shown in Figs 1 and 2, respectively. The intervention procedure is shown in Fig 3.

**Inclusion criteria.** The inclusion criteria are as follows: (1) aged 18–60 (either sex); (2) BMI of 18.5–23.9 kg/m$^2$; (3) no history of lower back pain; (4) normal range of lumbar mobility (including flexion from 75–90 degrees, extension of 30 degrees, lateral bending from 20–35 degrees, and unilateral rotation of 90 degrees); (5) willing to participate in the trial and have signed the informed consent form.

**Exclusion criteria.** The exclusion criteria are as follows: (1) participants with serious heart, liver, kidney, or hematological diseases; (2) women in menses, pregnancy, or lactation; (3) those unable to complete the imaging in the prone position (approximately 40 minutes); (4) those with cognitive impairments; (5) those with a history of lower back trauma or whiplash within a week; (6) those with skin diseases or skin lesions, sensory impairments, scarring, or neoplasms at the test site; (7) those with metal allergies; and (8) those who have participated in other clinical trials that may affect the results of the study within the last three weeks.

## Temperature detection and measurement

The FLIR E53 camera (FLIR Systems, Inc., Wilsonville, OR, USA) will be used to obtain thermal images, with a temperature sensitivity of 0.04°C, allowing for precise temperature

| | STUDY PERIOD | | | | | | | | | | | | |
|---|---|---|---|---|---|---|---|---|---|---|---|---|---|
| | | | | **Intervention** | | | | | | | **Post-intervention** | | |
| | **Enrolment** | **Allocation** | **Baseline** | **Duration (min)** | | | | | | | **After (min)** | | |
| **TIMEPOINT** | | | | 0 | 5 | 10 | 15 | 20 | 25 | 30 | 0 | 5 | 10 |
| **ENROLMENT:** | | | | | | | | | | | | | |
| Eligibility screen | × | | | | | | | | | | | | |
| Informed consent | × | | | | | | | | | | | | |
| General information | × | | | | | | | | | | | | |
| Allocation | | × | | | | | | | | | | | |
| **INTERVENTIONS:** | | | | | | | | | | | | | |
| Acupuncture | | | | ◆ | | | | | | ◆ | | | |
| Moxibustion | | | | ◆ | | | | | | ◆ | | | |
| **ASSESSMENTS:** | | | | | | | | | | | | | |
| Infrared thermal imaging (lumbar region) | | | × | | × | × | × | × | × | × | | | |
| Temperature recording (acupoint) | | | ◆ | | | | | | | | | | ◆ |
| Warming sensation questionnaire | | | | | | | | | | | × | | |
| **Adverse events** | | | | ◆ | | | | | | | | | ◆ |
| **Reasons for drop-out and withdrawals** | ◆ | | | | | | | | | | | | ◆ |

**Fig 2. Standard Protocol Items: Recommendations for Interventional Trials (SPIRIT) figure: Schedule of enrolment, interventions, and assessments.** In acupuncture group, twirling manipulation will be performed for 30 seconds every 10 minutes, with a total of two operations.

detection. The camera will be positioned perpendicular to the observation area, including the premarked area and acupoints (Figs 4 and 5), at a distance of 80 cm.

Each participant will be evaluated in a room with a constant temperature of 26°C and 40%-60% humidity. Thermal images will be captured at baseline (5 minutes before intervention), during intervention (0, 5, 10, 15, 20, 25, and 30 minutes), and postintervention (10 minutes after intervention completion), with all other irrelevant heat sources turned off.

Two observers will analyze the thermal images using the FLIR Tools application, with the lumbar region and acupoints predefined and outlined in the normal VGA photograph. The average and maximum temperatures of the selected area will be calculated using the application. The change in temperature over time will be compared between the different time points, and interobserver reliability will be calculated.

## Interventions

Considering the comparable anatomical features of BL40 and LU5 (both are located in the medial depression of the joint and have rich vascular and neural distribution in the local area) and the differences in the meridian pathways (the bladder meridian passes through the lumbar

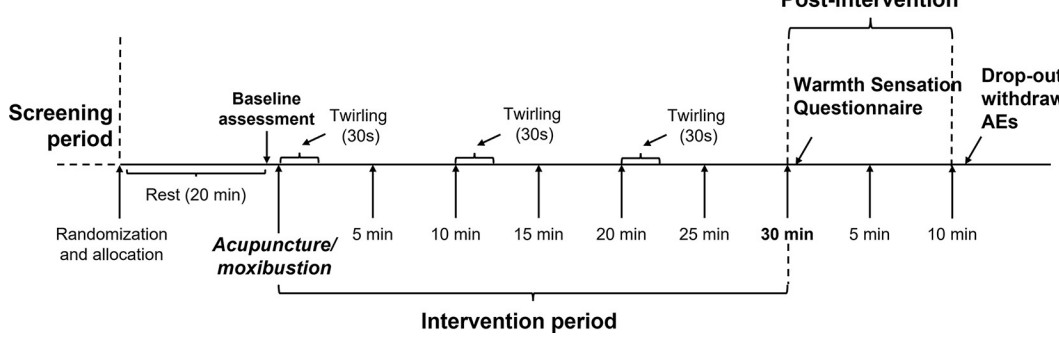

**Fig 3. Procedure of interventions and data collection time points.**

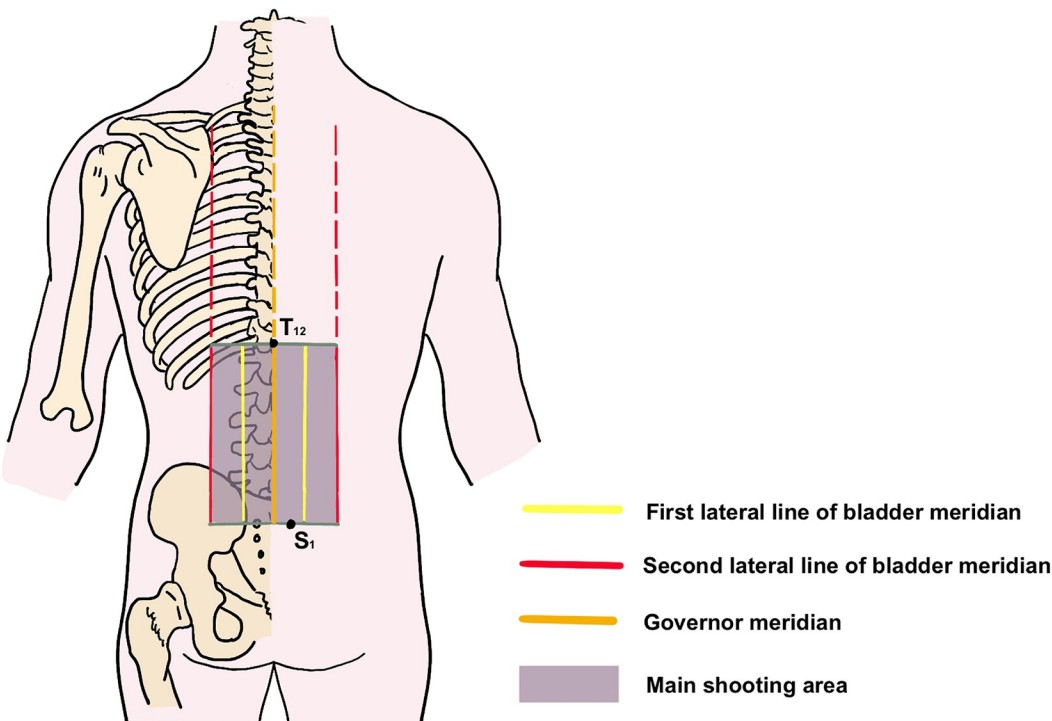

**Fig 4. Diagram of the lumbar area.** The purple rectangular area is enclosed by the second lateral line of the bladder meridian and a horizontal line extending from below the spinous process of the $T_{12}$ to the $S_1$ vertebra. The yellow line means the first lateral line of bladder meridian, the red line means the second lateral line of bladder meridian, the orange line means the governor meridian and the purple rectangle means the lumbar area.

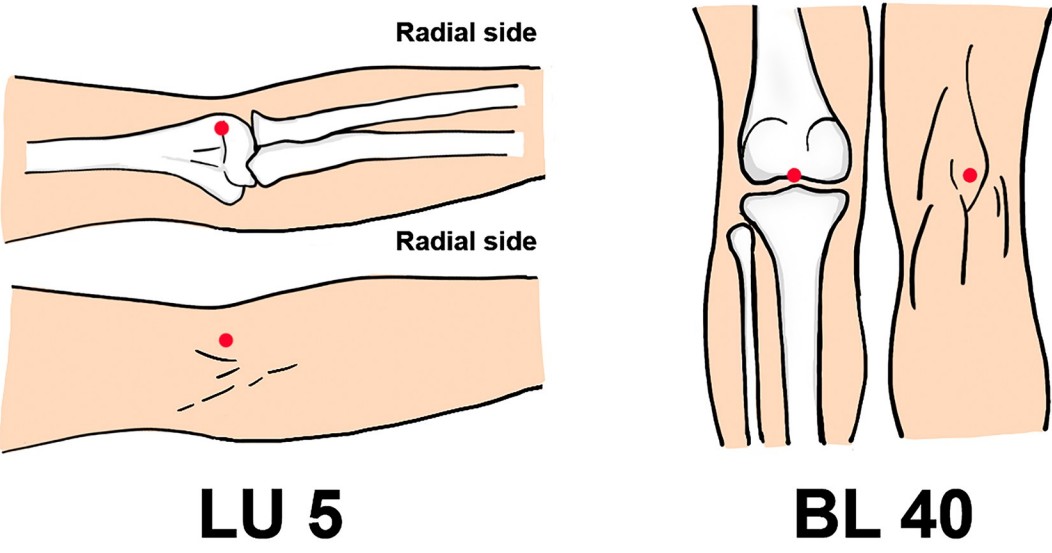

**Fig 5. Diagram of the acupoints.**

**Table 1. Acupuncture points selected in this protocol.**

| Acupuncture points | Location |
|---|---|
| Bilateral Chize (LU5) | On the transverse elbow stripe, in the depression of the radial border of the biceps tendon. |
| Bilateral Weizhong (BL40) | Posterior knee area, the midpoint of the transverse popliteal line. |

**Table 2. Groups and intervention allocation.**

| Groups | Interventions |
|---|---|
| Acupuncture BL40 group (Acu-BL40) | acupuncture at Weizhong (BL40) |
| Acupuncture LU5 group (Acu-LU5) | acupuncture at Chize (LU5) |
| Moxibustion BL40 group (Mox-BL40) | moxibustion at Weizhong (BL40) |
| Moxibustion LU5 group (Mox-LU5) | moxibustion at Chize (LU5) |

area while the lung meridian does not), we chose to use LU5 as the control. The specific acupoint locations are delineated in Table 1 and Fig 6. The acupuncture and moxibustion interventions will be conducted by licensed acupuncturists with a minimum of two years of experience. Each participant will complete a single 30-minute session of either acupuncture or moxibustion treatment. If any adverse events (AEs) occur during treatment, needles and moxa will be immediately removed. The intervention of the treatment group is shown in Table 2.

## Acupuncture

Participants randomly assigned to the acupuncture group will receive manual needling intervention at bilateral BL40 or LU5. The participants will be in a prone position, and after routine disinfection of the skin at the acupoint area with 75% alcohol, the needles (length: 40 mm, diameter: 0.25 mm; Hwato, Suzhou, China) will be inserted to a depth of 15–20 mm at a 90˚ angle. Then, the needle will be evenly lifted, thrusted in a range of 3–5 mm and twirled at a rotation angle of 90˚-180˚ with a frequency of 60–90 times/min. If subjects report a sensation

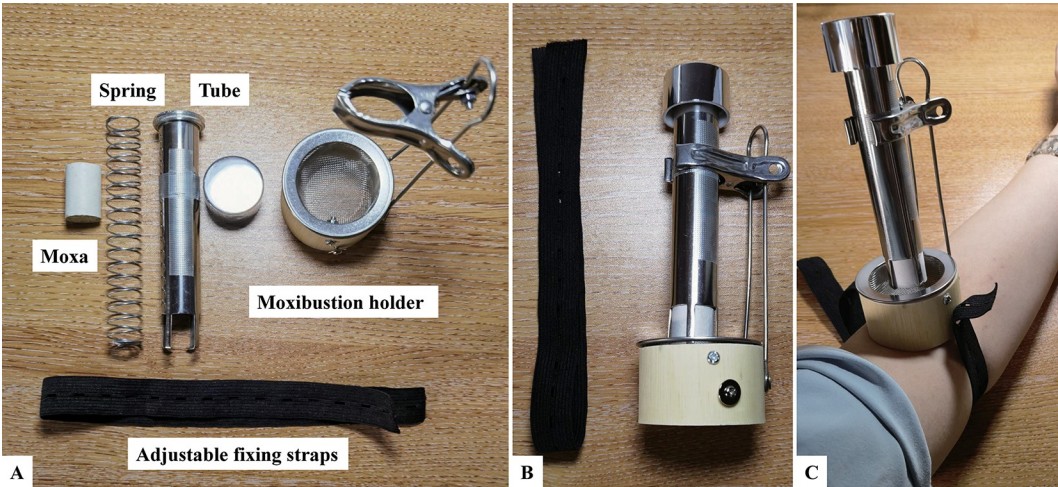

**Fig 6. Moxibustion device.** A, the components of the moxibustion device (including a moxibustion holder, a tube, a spring and moxa); B, the Assembled device; C, Adjustable fixing straps to hold the device in place.

of "Deqi [9]" (soreness, numbness, distention, or heaviness), the needle will be twirled for another 30 seconds. The total time of acupuncture will last for 30 minutes. The KSE36A0R thermometer will be placed near the acupoint to record the temperature. During the needle retention period, twirling manipulation will be performed for 30 seconds every 10 minutes, with a total of two operations.

**Moxibustion.** The participants in the moxibustion group will receive suspended moxibustion intervention at BL40 or LU5. Participants will assume the prone position, and the temperature sensor of the KSE36A0R thermometer will be placed on the acupoint before the intervention. The temperature alarm will be set at 42°C in the temperature recorder (which does not cause burning pain [10]). The suspension moxibustion apparatus (equipped with an automatic moxa feeder to keep the distance between the burning end of the moxa stick and the skin constant; Fig 6B and 6C) will be placed and fixed, and a moxa stick (18 mm in diameter and 60 mm in length, Han Yi, Nanyang, China) will be lit and placed in the automatic feeder and fixed on the suspension moxibustion apparatus. The distance between the burning end of the moxa stick and the skin will be set at 3 cm (safe moxibustion distance [11]), and the intervention duration will be 30 minutes.

When the participant feels a burning sensation or alarm sounds, the moxa stick will be moved upward by 0.5 cm, and after one minute, the participant will be asked again whether the burning sensation still exists. If so, the moxa stick was raised by another 0.5 cm until the burning sensation disappeared.

For better observation of temperature changes in the acupoints, a KSE36A0R thermometer (Keshun Instrument Co., Ltd., Ningbo, China) will be placed near the acupoint to record the temperature.

## Outcome measurement

We will use the FLIR E53 camera to record the infrared parameters of the lumbar region (a geometric shape bounded by the second lateral line of the bladder and the horizontal lines of $T_{12}$ and $S_1$ on both sides, as shown in Fig 4) at baseline, during and after the intervention at various time points to observe the thermal effects. Subsequently, participants will be requested to complete a questionnaire evaluating the presence of warmth sensation in their waist and the intensity of thermal perception after the trial.

**Baseline.** We will collect baseline information of the subjects, such as sex, age, weight, height, recent frequency(weekly) of lower back discomfort in the form of interviews and questionnaires, and whether they have received acupuncture/moxibustion related treatments. Prior to the intervention, we will use FLIR E53 to record the relevant infrared thermal parameters of the lumbar region (Fig 4).

**Primary outcome.** The primary outcome is mean temperature in the lumbar region (Fig 4) at the last minute of the intervention period.

## Secondary outcomes

1. Maximum temperature in the lumbar region (Fig 4) at the last minute of the intervention period.

2. Mean temperature in the lumbar region (Fig 4) at other time points during the intervention period (Fig 3).

3. Mean temperature in the bilateral first and second lateral line of the bladder meridian in the lumbar region (Fig 4) at the last minute of the intervention period.

4. Mean temperature in the lumbar region (Fig 4) at specific time points after the intervention period (Fig 3).

5. Warming sensation questionnaire at the end of the trial, including the sensation of warmth (secondary classification data with yes or no) and its intensity (NRS score from 0 to 10).

## Adverse events and safety

When they are properly administered, acupuncture and moxibustion are generally regarded as safe procedures. Nonetheless, this study aims to comprehensively document all AEs attributed to acupuncture and moxibustion in case report forms (CRFs). Such AEs encompass a range of issues, including broken needles, syncope resulting from needling or moxibustion, prolonged postneedling pain exceeding two hours, localized infection, hematoma, bleeding, burns, and other effects that may arise from acupuncture and moxibustion, such as fatigue, headache, insomnia, and dizziness. In cases of severe AEs, the attending physician will provide immediate clinical treatment, and the events should be reported to the principal investigator and the Institutional Review Board.

## Data collection and management

Following receipt of consent, a designated research assistant (RA), who is blinded to treatment assignment and has received standardized questionnaire administration training, will enter demographic and baseline characteristic data into CRFs. These data will include sex, age, weight, height, frequency of recent low back discomfort (measured weekly), acupuncture/moxibustion treatment history, and baseline temperature of the lumbar region (Fig 4).

Data collection will be restricted to members of the investigative team, who are obligated to maintain confidentiality and use the information solely for scientific research purposes. Data will be recorded in both electronic and paper formats and uploaded to an online data management team (ResMan research manager repository, supported by West China Hospital, Sichuan University, China, http://www.medresman.org), which will oversee the trial process. To ensure data confidentiality, only the investigators will be granted access via usernames and passwords. Statistical analyses will be conducted independently by two experimenters.

## Statistical analysis

Data analysis will be performed using Microsoft Excel 2016 and IBM SPSS Statistics 23.0 (SPSS Inc., Chicago, IL, United States). Data will be presented as the means and standard deviations, medians and interquartile ranges, or numbers and percentages, depending on the data type and distribution. Normality will be assessed using the Shapiro–Wilk test. Continuous data (nonnormally distributed data will be adjusted by appropriate methods), such as age, body mass index, temperature, and thermal intensity, will be assessed using independent samples t tests or analysis of variance (ANOVA). Categorical data, such as gender and sense of warmth, will be assessed using the chi-squared test or Fisher's exact test.

For the primary outcome (mean temperature in the lumbar region at the last minute of the intervention period), two-way analysis of variance (two-way ANOVA) will be used with a full factorial model that includes the main effect of acupoints and interventions and the interaction effect. If there is no interaction effect, simple effects of acupoints and interventions will be analyzed. The model will be adjusted by including baseline temperature as a covariate. Subgroup analyses of the primary outcome will be conducted according to the warming sensation questionnaire (YES vs. NO). For secondary outcomes, such as the mean temperature in the lumbar

region, maximum temperature in the lumbar region, and mean temperature in the bilateral first and second lateral line of the Bladder meridian, a general linear mixed effects model will be fitted. The main independent variables will be treatment group, time, gender, experience of acupuncture or moxibustion, body mass index, warming sensation questionnaire, and baseline temperature as covariates.

Data will be analyzed using the intention-to-treat (ITT) principle, and sensitivity analyses of the primary outcome will be conducted using the per protocol set (PPS) along with factorial analysis. Missing data will be imputed using multiple imputation methods. A two-sided P value of 0.05 will be considered statistically significant in the analysis.

### Ethics approval and consent to participate

This research was carried out in accordance with the Helsinki Declaration. The trial is listed on ClinicalTrials.gov (NCT05665426). The trial was approved by the Ethics Committee of Zhejiang Chinese Medicinal University's Third Affiliated Hospital (ZSLL-ZN-2022-026-01), and detailed information can be found in (S1 File). Before enrolling in this trial, each participant will be given a consent form outlining the study procedures, potential risks, and rights. Participants will sign an informed consent form and agree to the publication of the study results in a peer-reviewed journal. Personal information about all trial participants will be kept private both during and after the trial. All participant data are accessible only to authorized researchers.

### Quality control

Acupuncture, methodology, and statistics experts will review and revise the protocol. All observations in clinical studies will be verified and confirmed multiple times to ensure the data's reliability and originality, and that all clinical study results and conclusions are derived from the original data.

The Ethics Board of The Third Affiliated Hospital of Zhejiang Chinese Medicinal University, whose members have no conflict of interest with this study, will oversee the research. The principal investigators will have access to all results and will have the final say on whether or not to end the study. They will also grant project members the right to publish papers containing the results of this trial.

## Discussion

LBP is a common clinical problem, with up to 84% of adults experiencing different degrees of LBP [12, 13]. However, due to the self-limiting nature of LBP and the side effects of Western medicine, most individuals prefer to seek nonmedical assistance [14]. Acupuncture and moxibustion are effective and widely used methods for treating lumbar disc herniation, and BL40 is one of the most commonly used acupoints for treating this condition [15–17]. Moreover, acupuncture and moxibustion share a common therapeutic approach of stimulating acupoints, while compared with acupuncture, moxibustion seems to be more acceptable and easier to operate by ordinary people. However, it is unclear whether there are differences in their effectiveness. Currently, few studies have compared the effects of acupuncture and moxibustion. To comprehensively analyze the characteristics of the BL40 acupoint, this trial adopts a 2×2 factorial design, which can simultaneously study the differences in thermal effects generated by different interventions (manual acupuncture or moxibustion) and different acupoints (BL40 or LU5).

The limitation of this protocol is the inability to blind participants and acupuncturists, and the intervention is a single session, lacking the study of cumulative effects. Regarding blinding,

considering the lack of knowledge among participants about meridians and the insufficient understanding of the therapeutic effects of various acupoints, to some extent, this study is also a form of blinding. Additionally, the result assessors and statistical analysts are also unaware of the intervention grouping, which greatly reduces potential bias and ensures the quality of this trial. With regard to the study of cumulative effects, since this trial involves healthy individuals as the study population, the research value of studying cumulative effects is limited, and we will conduct further experimental studies on patients with LBP.

In summary, the results of this trial will indirectly observe changes in skin microcirculation through infrared thermography to demonstrate the differences in the therapeutic effects of BL40 and LU5 on the lower back and the differences in the skin microcirculation therapeutic effects of acupuncture and moxibustion. The results of this study are expected to provide new insights and evidence for the traditional Chinese medicine theory of "Yao Bei Wei Zhong Qiu" and fill the research gap in this field.

## Supporting information

**S1 Checklist. SPIRIT 2013 checklist: Recommended items to address in a clinical trial protocol and related documents\*.**
(DOC)

**S1 File. Approval letter and protocol.**
(DOCX)

**S2 File.**
(DOCX)

## Acknowledgments

The authors would like to appreciate the participation of the included subjects who will be involved in this trial.

## Author Contributions

**Conceptualization:** Zhengyi Lyu, Yi Liang.

**Investigation:** Qiongying Shen.

**Methodology:** Siyi Zheng, Qiongying Shen, Zhengyi Lyu, Xiaoxiao Huang, Xiaoshuai Yu, Yi Liang, Jianqiao Fang.

**Project administration:** Yi Liang.

**Resources:** Yiyue Liu.

**Software:** Siyi Zheng, Shuxin Tian, Yiyue Liu, Wei Pan.

**Validation:** Na Nie, Yi Liang, Jianqiao Fang.

**Writing – original draft:** Siyi Zheng.

**Writing – review & editing:** Shuxin Tian, Yi Liang.

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
