## [Decision Letter · Decision Letter 0]

4 Jul 2023

PONE-D-23-12555Effect of acupuncture or moxibustion at Acupoints Weizhong or Chize on the change in lumbar temperature in healthy adults: a study protocol for a randomized controlled trial with a 2 × 2 factorial designPLOS ONE

Dear Dr. Liang,

Thank you for submitting your manuscript to PLOS ONE. After careful consideration, we feel that it has merit but does not fully meet PLOS ONE’s publication criteria as it currently stands. Therefore, we invite you to submit a revised version of the manuscript that addresses the points raised during the review process.

We look forward to receiving your revised manuscript.

Kind regards,

Boram Lee

Academic Editor

PLOS ONE

Journal Requirements:

   "This study is funded by The National Key Research and Development Program of China (No. 2018YFC1704600)."

Additional Editor Comments:

1. The rationale should be clearer about what it means to evaluate the effects of acupuncture and moxibustion separately.

2. It is needed that further explanation of why and how important increasing microcirculation is.

3. Further explanation of why and how important increasing microcirculation is.

4. Your manuscript differs in some respects from the registry (clinicaltrials.gov). A clear explanation should be given as to why it differs from the registered protocol. This can lead to serious reporting bias.

5. It is necessary to reexamine whether the SPIRIT checklist was properly followed. Please review whether N/A was entered in an inappropriate place.

Reviewers' comments:

Reviewer's Responses to Questions

**Comments to the Author**

1. Does the manuscript provide a valid rationale for the proposed study, with clearly identified and justified research questions?

Reviewer #1: Partly

Reviewer #2: Partly

Reviewer #3: Yes

2. Is the protocol technically sound and planned in a manner that will lead to a meaningful outcome and allow testing the stated hypotheses?

Reviewer #1: Yes

Reviewer #2: Yes

Reviewer #3: Yes

3. Is the methodology feasible and described in sufficient detail to allow the work to be replicable?

Reviewer #1: Yes

Reviewer #2: Yes

Reviewer #3: Yes

4. Have the authors described where all data underlying the findings will be made available when the study is complete?

Reviewer #1: Yes

Reviewer #2: Yes

Reviewer #3: Yes

5. Is the manuscript presented in an intelligible fashion and written in standard English?

Reviewer #1: Yes

Reviewer #2: Yes

Reviewer #3: Yes

6. Review Comments to the Author

You may also provide optional suggestions and comments to authors that they might find helpful in planning their study.

Reviewer #1: This study plan to detect the dynamic changes of skin temperature in a low back area after acupuncture and moxibustion at BL 40 Weizhong point and LU 5 Chize point. It can provide some evidence of relationship between BL 40 and "waist and back".

1. Another group may be added, eg. BL 39 weiyang or BL 37 Yinmen, other Bladder points, and a sham point group should be added, close to BL 40, while not on Bladder meridian.

2.Inclusion criteria, Age ≥18 year, no upper limit now.

3. The names of the mentioned acupuncture points should be presented properly. For example, Weizhong point and Chize point, should add BL40 and LU 5.

4. The language should be further modified. Some grammatical errors should be revised.

Reviewer #2: In this study, authors designed the protocol of assess the effect of acupuncture or moxibustion at Acupoints Weizhong or Chize on the change in lumbar temperature in healthy adults:. It is interesting. However, I have some comments:

①How to control the temperature during moxibustion treatment?

②CONSORT checklist should be added in the manuscript.

③Why LU5 was selected as the control ?

Reviewer #3: Line 17: Change to "will compare".

Line 28: Suggest to plan for a per-protocol analysis as a sensitivity analysis; the intent-to-treat analysis should be the primary analysis.

Line 86: Change to "although only about 40% of participants can feel the warmth at the lumbar region".

Line 87: Change to "decided".

Line 214: Note that the ability to perform analysis is quite diminished if using the Kruskal-Wallis test, in comparison with the linear model. Therefore, it will be best if the linear model can be applied even if there is some lack of fit in some of the comparisons. Note that the estimates are still unbiased, it is only the statistical significance calculation that is generally affected, and this might be rescued via transformation in such cases.

Line 217: What method will be used for post hoc evaluations of interaction effects? Simple effects? Pairwise comparisons? And, what correction will be used for multiple testing, if any?

Note that a linear mixed effect model can accommodate the repeated measures structure that is planned, with multiple time points measured, as well as accommodate covariate information for evaluation of other factors. With the number of subjects planned, it should be feasible to perform such an analysis using an unstructured correlation matrix for the R-side. It can even handle other outcome types via the generalized linear mixed effect model extension. This framework provides a more unified setting to perform all of the envisioned analyses.

7. PLOS authors have the option to publish the peer review history of their article (what does this mean?). If published, this will include your full peer review and any attached files.

Reviewer #1: No

Reviewer #2: **Yes: **Wei Huang

Reviewer #3: No

---

## [Author Response · Author response to Decision Letter 0]

27 Jul 2023

Response to Additional Editor Comments:

Comments 1. The rationale should be clearer about what it means to evaluate the effects of acupuncture and moxibustion separately.

Response: Thank you for your suggestion. To provide a clearer rationale for evaluating the effects of acupuncture and moxibustion separately, the sentence “Although both acupuncture and moxibustion are extensively used in clinical treatment, moxibustion is less painful, more user-friendly and patient-acceptable” had been supplemented in the Background of the manuscript (Lines 56-57, 259-260).

Comments 2-3. It is needed that further explanation of why and how important increasing microcirculation is. Further explanation of why and how important increasing microcirculation is. 

Response: Thank you for your kindly reminding. To provide further explanation on the importance of increasing microcirculation, we have revised the manuscript as following: “In clinic, the change of skin microcirculation can be visualized directly by the laser Doppler technique (skin blood flow) and indirectly by the infrared thermography (body surface temperature). Infrared thermographic imaging is also a reliable and easy-to-handle tool to distinguish between different interventions of needling [4]. Acupuncture at BL40 can relive pain in LBP patients, and the increase skin microcirculation can also be observed in the lumbar region of LBP patients who treated by needling at BL40 [5, 6], which suggest that the increase of skin microcirculation might be associated with the relief of the LBP. What’s more, the similar phenomenon (increase of skin microcirculation after needling at BL40) was also occur in the healthy individuals [7, 8].” (Line 49-56)

Comments 4. Your manuscript differs in some respects from the registry (clinicaltrials.gov). A clear explanation should be given as to why it differs from the registered protocol. This can lead to serious reporting bias.

Response: Thank you for your comments. Actually, the protocol had been modified according to experts’ opinion and clinical practice, which demonstrated at the manuscript. However, the registered protocol (clinicaltrials.gov) was the original version and did not update simultaneously. That was the main reason which led to the difference. And now we have already updated the newest protocol in clinicaltrials. 

The details of the modified protocol are showed as below:

 (https://classic.clinicaltrials.gov/ct2/history/NCT05665426?B=8&A=7&C=merged#StudyPageTop): 

 PS:corresponding images are in "Response to Reviewer" file

Comments 5. It is necessary to reexamine whether the SPIRIT checklist was properly followed. Please review whether N/A was entered in an inappropriate place.

Response: Thank you for your kindly reminding of the SPIRIT checklist. Upon reexamination, we found a few instances where "N/A" was entered inappropriately. We have corrected these parts and colored them in red.

Reply to Reviewers' comments:

Reviewer #1: This study plan to detect the dynamic changes of skin temperature in a low back area after acupuncture and moxibustion at BL 40 Weizhong point and LU 5 Chize point. It can provide some evidence of relationship between BL 40 and "waist and back".

Comments 1. Another group may be added, eg. BL 39 weiyang or BL 37 Yinmen, other Bladder points, and a sham point group should be added, close to BL 40, while not on Bladder meridian.

Response: Thank you for your professional advice. We strongly agree that adding another group such as BL39, BL37 and a sham point close to BL40 as control would improve the value of this study. However, considering that BL39, BL37 and BL40 are all located on the SAME meridian, it may be more difficult to find their differences. What’s more, considering the thermal radiation range of moxibustion, it is difficult to find a sham point which close to BL40 but is not affected by thermal radiation. Therefore, LU5 was selected as control in this study. And in some way, LU5 can also be considered as the “sham point group” because of the difference meridian pathway (the Bladder meridian passes through the lumbar area while the Lung meridian does not). BL39, BL37 and sham point close to BL40 will be considered in our future studies.

Comments 2. Inclusion criteria, Age ≥18 year, no upper limit now.

Response: Thank you for kindly reminder. We have modified the inclusion criteria of age as 18-60 years to make the inclusion criteria more completed. And the information has been updated on the website (clinicaltrials.gov).

Comments 3. The names of the mentioned acupuncture points should be presented properly. For example, Weizhong point and Chize point, should add BL40 and LU 5.

Response: Thank you for kindly reminder. We have revised the manuscript and added "BL40" for Weizhong point and "LU5" for Chize point to ensure clarity and accuracy in the presentation of the acupuncture points. We appreciate your attention to detail and your valuable input in improving the quality of our manuscript.

Comments 4. The language should be further modified. Some grammatical errors should be revised.

Response: Thank you very much for the comments. We have carefully reviewed the language and made the necessary revisions to improve the clarity and accuracy of the text.

Reviewer #2: In this study, authors designed the protocol of assess the effect of acupuncture or moxibustion at Acupoints Weizhong or Chize on the change in lumbar temperature in healthy adults:. It is interesting. However, I have some comments:

Comments 1. How to control the temperature during moxibustion treatment?

Response: Thank you so much for this kindly reminder. We have moved this section from “Temperature detection and measurement” to “Moxibustion” to make it clearer. See as follows: “For better observation of temperature changes in the acupoints, KSE36A0R thermometer (Keshun Instrument Co., Ltd., Ningbo, China) will be placed near the acupoint to record the temperature.” (Line 143-144)

Comments 2. CONSORT checklist should be added in the manuscript.

Response: Thank you for your valuable suggestion. We had submitted the SPIRIT checklist as supplementary materials. And in our future RCT report, we will ensure to include the CONSORT checklist to adhere to the appropriate reporting guidelines.

Comments 3. Why LU5 was selected as the control?

Response: Thank you for your comments. Several statements that we made were more ambiguous than intended, and we have adjusted to the text to be clear. We have adjusted the sentence as following: “Considering the comparable anatomical features of BL40 and LU5 (Both located in the medial depression of the joint and have rich vascular and neural distribution in the local area) and the difference meridian pathway (the Bladder meridian passes through the lumbar area while the Lung meridian does not), we have chosen to use LU5 as the control.” (LINE 137-139)

Reviewer #3:

Line 17: Change to "will compare".

Line 86: Change to "although only about 40% of participants can feel the warmth at the lumbar region".

Line 87: Change to "decided".

Response: Thank you for your kindly reminder. We have made the necessary modifications to the manuscript to address the issues you pointed out.

Line 28: Suggest to plan for a per-protocol analysis as a sensitivity analysis; the intent-to-treat analysis should be the primary analysis.

Line 214: Note that the ability to perform analysis is quite diminished if using the Kruskal-Wallis test, in comparison with the linear model. Therefore, it will be best if the linear model can be applied even if there is some lack of fit in some of the comparisons. Note that the estimates are still unbiased, it is only the statistical significance calculation that is generally affected, and this might be rescued via transformation in such cases.

Line 217: What method will be used for post hoc evaluations of interaction effects? Simple effects? Pairwise comparisons? And, what correction will be used for multiple testing, if any?

Note that a linear mixed effect model can accommodate the repeated measures structure that is planned, with multiple time points measured, as well as accommodate covariate information for evaluation of other factors. With the number of subjects planned, it should be feasible to perform such an analysis using an unstructured correlation matrix for the R-side. It can even handle other outcome types via the generalized linear mixed effect model extension. This framework provides a more unified setting to perform all of the envisioned analyses.

Response: Thank you for your highly professional comments. We agree that the generalized linear mixed effect model extension is a better choice for our statistical analysis, and we have reconsidered the plan of statistical analysis, and rewrote this part (Line 27-28, 215-235).

Primary changes are as follows:

1. According to your advice, we have included a per-protocol analysis as a sensitivity analysis and put the intent-to-treat analysis as the primary analysis.

2. For the primary outcome measures, we plan to use factorial analysis of variance to conduct the analysis. 

3. As for the secondary outcome measures involving repeated measurements, we will use the generalized linear mixed effect model to analyze the data. This approach allows us to accommodate the repeated measure structure, multiple time points, and covariate information, providing a more unified and comprehensive framework for all the envisioned analyses.

We are confident that the revised manuscript now meets the high standards set by the journal. We believe that these revisions have strengthened the clarity, methodology, and overall contribution of our study. We hope that you will find our responses and revisions satisfactory.

Once again, we sincerely appreciate your thorough review and constructive feedback. We would like to express our gratitude for your time and expertise in evaluating our work. We believe that your inputs have significantly improved the quality of our manuscript.

We kindly request that you reconsider our manuscript for publication in Plos one. We are confident that the revised version now meets all the requirements and would make a valuable contribution to the scientific community.

Thank you for your attention and consideration.

Sincerely,

Yi Liang

---

## [Decision Letter · Decision Letter 1]

22 Aug 2023

PONE-D-23-12555R1Effect of acupuncture or moxibustion at Acupoints Weizhong (BL40) or Chize (LU5) on the change in lumbar temperature in healthy adults: a study protocol for a randomized controlled trial with a 2 × 2 factorial designPLOS ONE

Dear Dr. Liang,

Thank you for submitting your manuscript to PLOS ONE. After careful consideration, we feel that it has merit but does not fully meet PLOS ONE’s publication criteria as it currently stands. Therefore, we invite you to submit a revised version of the manuscript that addresses the points raised during the review process.

We look forward to receiving your revised manuscript.

Kind regards,

Boram Lee

Academic Editor

PLOS ONE

Journal Requirements:

Additional Editor Comments:

**English must be improved before publishing, and authors are encouraged to submit English editorial certificate.**

Reviewers' comments:

Reviewer's Responses to Questions

**Comments to the Author**

1. Does the manuscript provide a valid rationale for the proposed study, with clearly identified and justified research questions?

Reviewer #1: Yes

Reviewer #3: Yes

2. Is the protocol technically sound and planned in a manner that will lead to a meaningful outcome and allow testing the stated hypotheses?

Reviewer #1: Yes

Reviewer #3: Yes

3. Is the methodology feasible and described in sufficient detail to allow the work to be replicable?

Reviewer #1: Yes

Reviewer #3: Yes

4. Have the authors described where all data underlying the findings will be made available when the study is complete?

Reviewer #1: Yes

Reviewer #3: Yes

5. Is the manuscript presented in an intelligible fashion and written in standard English?

Reviewer #1: Yes

Reviewer #3: Yes

6. Review Comments to the Author

You may also provide optional suggestions and comments to authors that they might find helpful in planning their study.

Reviewer #1: The authors have adjusted the text, the revisions have strengthened the clarity and methodology. The protocol technically sound and will lead to a meaningful outcome. It can provide evidence of relationship between BL 40 and "waist and back".

Reviewer #3: The authors have responded constructively to all of the review comments.

It would be best if the authors exactly specify the statistical analysis of the primary endpoint. The analysis of variance is a rather general term for the overall methodology. The authors should specify what exactly they will test. One possibility is to test for any effect (inclusive of main effect and/or interaction effect). This could be performed via an omnibus test of the full model versus a restricted model that does not include treatment (or any interaction involving treatment).

7. PLOS authors have the option to publish the peer review history of their article (what does this mean?). If published, this will include your full peer review and any attached files.

Reviewer #1: No

Reviewer #3: No

---

## [Author Response · Author response to Decision Letter 1]

30 Aug 2023

Reply to Reviewers' comments:

Reviewer #1: The authors have adjusted the text, the revisions have strengthened the clarity and methodology. The protocol technically sound and will lead to a meaningful outcome. It can provide evidence of relationship between BL 40 and "waist and back".

Response: Thank you for your approval.

Reviewer #3: 

The authors have responded constructively to all of the review comments.

It would be best if the authors exactly specify the statistical analysis of the primary endpoint. The analysis of variance is a rather general term for the overall methodology. The authors should specify what exactly they will test. One possibility is to test for any effect (inclusive of main effect and/or interaction effect). This could be performed via an omnibus test of the full model versus a restricted model that does not include treatment (or any interaction involving treatment).

Response: Thank you for your professional advice. We have operationalized the statistical approach as Two-Way analysis of variance, and we have refined the statistical strategy accordingly. we have revised the manuscript as following: “For the primary outcome (mean temperature in the lumbar region at the last minute of the intervention period), two-way analysis of variance (two-way ANOVA) will be used with a full factorial model that includes the main effect of acupoints and interventions and the interaction effect. If there is no interaction effect, simple effects of acupoints and interventions will be analyzed. The model will be adjusted by including baseline temperature as a covariate.”

We are confident that the revised manuscript now meets the high standards set by the journal. We believe that these revisions have strengthened the clarity, methodology, and overall contribution of our study. We hope that you will find our responses and revisions satisfactory.

---

## [Editor Report · Decision Letter 2]

1 Sep 2023

Effect of acupuncture or moxibustion at Acupoints Weizhong (BL40) or Chize (LU5) on the change in lumbar temperature in healthy adults: a study protocol for a randomized controlled trial with a 2 × 2 factorial design

PONE-D-23-12555R2

Dear Dr. Liang,

We’re pleased to inform you that your manuscript has been judged scientifically suitable for publication and will be formally accepted for publication once it meets all outstanding technical requirements.

Kind regards,

Boram Lee

Academic Editor

PLOS ONE
---

## [Editor Report · Acceptance letter]

19 Oct 2023

PONE-D-23-12555R2 

Effect of acupuncture or moxibustion at Acupoints Weizhong (BL40) or Chize (LU5) on the change in lumbar temperature in healthy adults: a study protocol for a randomized controlled trial with a 2 × 2 factorial design 

Dear Dr. Liang:

I'm pleased to inform you that your manuscript has been deemed suitable for publication in PLOS ONE. Congratulations! Your manuscript is now with our production department. 

Kind regards, 

on behalf of

Dr. Boram Lee 

Academic Editor

PLOS ONE